# Additional Health Benefits Observed following a Nature Walk Compared to a Green Urban Walk in Healthy Females

Yvanna Todorova [1,*], Izzy Wellings [1], Holly Thompson [1], Asya Barutcu [1], Lewis James [1], Nicolette Bishop [1], Emma O'Donnell [1], Colin Shaw [2,†] and Daniel P. Longman [1,3,†]

[1] School of Sport, Exercise, and Health Science, Loughborough University, Loughborough LE11 3TU, UK; i.g.wellings@lboro.ac.uk (I.W.); h.r.thompson-20@student.lboro.ac.uk (H.T.); a.barutcu3@lboro.ac.uk (A.B.); l.james@lboro.ac.uk (L.J.); n.c.bishop@lboro.ac.uk (N.B.); e.odonnell@lboro.ac.uk (E.O.); d.longman@lboro.ac.uk (D.P.L.)

[2] Department of Evolutionary Anthropology, University of Zurich, 8057 Zurich, Switzerland; colin.shaw@uzh.ch

[3] Institute of Sport Science (ISSUL), University of Lausanne, 1015 Lausanne, Switzerland

[*] Correspondence: y.todorova@lboro.ac.uk

[†] This symbol denotes joint last author.

**Abstract:** Chronic stress and obesity are major public health concerns and represent significant risk factors for a plethora of non-communicable diseases. Physical exercise represents a valuable health intervention in both cases, providing benefits for mental and physical health, as well as appetite regulation. While the emerging field of 'green exercise' suggests that the presence of nature may amplify the benefits of exercise, the quality of evidence has been questioned. To address this, we recruited 22 healthy females to complete a crossover randomised trial comprising a 75 min walk in both a forest and urban area, separated by 2–7 days. Markers of mood (Profile of Mood States), stress (sympathetic-adreno-medullar [resting heart rate, blood pressure] and hypothalamic–pituitary axis activation [salivary cortisol]) and eating behaviour (energy intake and salivary ghrelin) were measured before and after each walk. While both walking interventions improved mood and reduced physiological stress, the nature intervention (but not the urban intervention) also led to further improvements in total mood disturbance, depression, confusion and esteem-related affect ($F_{(1,21)} \geq 4.98$, $p \leq 0.037$). Salivary ghrelin ($F_{(20)} = 0.229$, $p = 0.637$) and energy intake ($t_{(20)} = -0.54$, $p = 0.60$) did not respond differently in the two environments. Overall, while walking improved mood and physiological stress in both environments, walking in a forested environment provided additional benefits for mood not seen following the urban walk.

**Keywords:** stress; forest bathing; mood; eating behaviour; walking; physical activity; salivary cortisol; green exercise

## 1. Introduction

The World Health Organisation describes stress as the 'health epidemic of the 21st Century' [1]. In 2018, the majority of UK adults (74%) were at some point overwhelmed by stress and unable to cope, with young adults being particularly affected (93%) [2]. Chronic stress can lead to a deterioration of both mental and physical health [3] and is considered a causative factor in the development of various long-term conditions, including depression and anxiety [2]. Stress can also lead to poorer cardiovascular [4], cardiometabolic [5], and musculoskeletal [6] health. In parallel, there is strong evidence that stress-induced immunosuppression is a key mechanism by which stress contributes to physical illness, either directly or indirectly via stress-induced behavioural changes [7].

In the short term, the stress response is advantageous because it prioritises the physiological functions necessary to preserve core function when challenged by a stressor, thereby promoting survival. Upon encountering a threat, the body responds by activating

the sympathetic–adrenal–medullary (SAM) and hypothalamic–pituitary–adrenal (HPA) axes [7]. Activation of the SAM axis enhances sympathetic nervous system (SNS) activity, which is responsible for the 'fight or flight' response [3]. In parallel, parasympathetic nervous system (PNS) activity, which acts to promote rest, recovery and food digestion, is decreased [8]. SNS activation leads to the dilation of blood vessels supplying the brain, heart and skeletal muscles, and an increase in heart rate, blood pressure and blood glucose in order to mobilise energy stores and shunt oxygen towards these essential tissues. Acute HPA axis activation generates a hormonal response to stress [9], leading to increased secretion of the glucocorticoid hormone cortisol by the adrenal glands. Cortisol mobilises glucose in the liver and fatty acids from adipose tissue and also attenuates the immune–inflammatory response, all of which helps to prioritise physiological functions related to immediate survival [9,10]. This stress response mechanism, although beneficial in an acute timescale when faced with an immediate threat, becomes harmful when chronically activated. Unfortunately, in contemporary industrialised societies, chronic activation of the stress response system has become widespread [3,11].

Chronic stress, in addition to being detrimental to various aspects of physiology and mental health [4–6,12], is linked with obesity. Alongside genetic and environmental factors [13], stress is understood to increase the risk of weight gain and obesity through cognitive, physiological, behavioural, and biochemical mechanisms. Central to each of these mechanistic responses are stress-induced increases in circulating cortisol and ghrelin, which may promote food intake and a preference for energy-dense sugary and fatty foods [14]. Over a period of months or years, this increased energy intake and fat storage can lead to weight gain and obesity. The social stigma surrounding obesity may itself act as an additional stressor that can lead to further weight gain [14]. Like stress, obesity represents a major contemporary health challenge, with 63% of the UK and 63% of the US adult population being either overweight (body mass index > 25 kg/m$^2$) or obese (body mass index > 30 kg/m$^2$ [15,16]. Obesity contributes to additional public health challenges and a decreased quality of life by increasing the risk of developing type 2 diabetes, coronary heart disease, select cancers, stroke and poor mental health [13].

### 1.1. Beneficial Effects of Physical Activity

The stress-relieving effects of physical activity are well documented [17,18] and positively influence both physical and mental health. Regular physical activity reduces the incidence of hypertension, cardiovascular disease, obesity, and depression, all of which are exacerbated by chronic stress [19,20]. Physical activity can also enhance immune function by reducing stress [21].

Physical activity can also act as a valuable intervention for the prevention and treatment of obesity through increased energy expenditure and improved metabolic fitness [22]. This is clinically significant, as a 10% weight loss in obese individuals causes a meaningful reduction in the risk of obesity-related comorbidities, while alleviating the symptoms of pre-existing conditions [23]. Even when physical activity does not result in weight loss, people with obesity still experience physiological benefits, including decreased insulin resistance and lowered blood pressure [24].

Unfortunately, chronic stress and obesity are rising and physical activity levels are declining [25]. In 2019, 33% of UK adults failed to meet the NHS physical activity guidelines, which was greater than the global figure of 28% [26,27]. While interventions designed to improve engagement with physical activity have had some success in improving health [17], levels of physical activity continue to fall [26].

### 1.2. Beneficial Effects of Exposure to Nature

There is a rapidly growing body of empirical evidence describing the stress-reducing effects of exposure to natural environments [28,29]. Whereas high levels of chronic stress are associated with urban living [30], exposure to natural environments has a relaxing effect on human physiology. Spending time in nature has been shown to stimulate PNS activity

and reduce SNS, SAM and HPA axis activity (reduce blood pressure, resting heart rate, and cortisol concentrations). These changes are accompanied by improved physical and mental health, including stabilised emotional affect, improved mood and, in some cases, enhanced immune function [31–34]. Furthermore, systematic reviews of greenspace access have also noted a lower incidence of obesity and obesity-related health indicators [35,36]. These effects appear to be particularly strong for females [37]. The COVID-19 pandemic is considered to have heightened positive associations between nature and wellbeing, as access to social activities was severely restricted in many countries [38,39]. This association seems to have been greatest for women, who appeared to benefit to a greater degree than men from green spaces [37].

Research investigating 'green exercise' (exercise undertaken in natural environments [40]) has considered whether exercising in nature confers benefits beyond those provided by physical activity in non-natural environments [40]. It has now been demonstrated that, compared against appropriate control conditions, green exercise can reduce physiological and psychological stress, increase feelings of social connectedness and improve mood [41] and cognitive performance [42]. Importantly, natural environments may alter an individual's experience of exercise. A 2013 review found that participants' perceived exertion was lower in natural settings despite achieving a faster self-selected jogging speed [43]. This reduction in perceived exertion, and associated boost in motivation, has the potential to increase public engagement with regular physical activity. However, there is limited support for the additional benefits of green exercise, with issues largely stemming from the low quality of evidence (small sample size and high risk of bias) currently available [44].

### 1.3. The Current Study

Chronic stress and obesity are major public health concerns and represent significant risk factors for a plethora of non-communicable diseases. Here, we apply an experimental approach to investigate whether the well-documented benefits of physical activity [19,20] are enhanced by exercising in a natural, as opposed to urban, environment.

## 2. Materials and Methods

### 2.1. Participants

Twenty-two females completed the study (age = 23.3 years, SD = 2.9 years; BMI = 22.6 kg/m$^2$, SD = 2.3 kg/m$^2$; percent fat-free mass = 72.4%, SD = 5.3%), nine of whom were using hormonal contraception. Female participants were recruited in an effort to balance the underrepresentation of females in nature-related and physiological research [43–49]. Potential participants were excluded if their BMI exceeded 30 kg/m$^2$, they were amenorrhoeic, hypertensive (blood pressure ≥140/90 mmHg), lactating, pregnant, smokers, excessive habitual drinkers (>14 units per week), or had diagnosed mood disorders or medical conditions that restricted exercise (e.g., asthma, diabetes). A power analysis was conducted using G*Power 3.1 [50] to determine the sample size required to test the difference between group means using $2 \times 2$ ANOVA. Previously reported nature-induced changes in cortisol [45] were used to identify a medium–large effect size [Cohen's dz = 0.6 [51]]. (Note: cortisol was used as there is no pre-existing data for measures of immune, cognitive or physical function). Ethical approval was granted by the Loughborough University Ethics Review Sub-Committee (Review Reference: 2021-3182-3623).

### 2.2. Intervention Protocol

A randomised crossover design was employed, with each participant attending two sessions. The sessions were scheduled between days one and seven of the menstrual cycle [52], when progesterone and oestrogen are lowest and have the least influence on energy intake [53]. Participants were asked to fast overnight, avoiding all food and drink, except water, prior to attending each session. Participants were randomly assigned to one of two intervention groups; in Group 1, participants completed the nature intervention first and the urban control intervention second, and vice versa in Group 2. Randomisation was

performed in clusters of 30 (15 in each intervention group) to ensure the equal distribution of groups. The nature walk took place in the Burleigh Woods on the Loughborough University Campus (Supplemental Materials). The green urban walk took place on Ashby Road, a highway near Loughborough (UK) town centre (Figure 1).

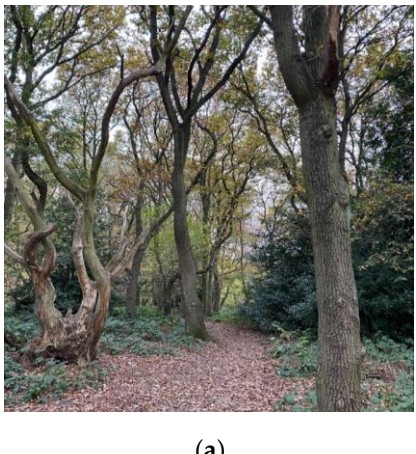

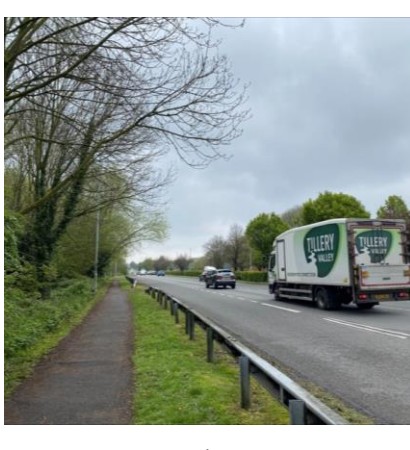

(**a**)  (**b**)

**Figure 1.** Photographs of the intervention locations. Photographs of nature (**a**) and green urban (**b**) intervention locations. The nature intervention was completed in the 8.5-hectare Burleigh Woods on the Loughborough University campus. Please see Supplemental Information S1 for further information. The green urban intervention took place along the dual carriageway (highway) Ashby Road (A512).

On testing days, participants arrived at the National Centre for Sport and Exercise Medicine laboratories at Loughborough University between 0800 and 1000. Upon arrival, they confirmed that they had completed the overnight fast and reported no experience of significant stress in the previous 24 h. Baseline measures of physiological stress [resting heart rate (RHR), blood pressure (BP), salivary cortisol and mood [Profile of Mood States (POMS)] were taken, as well as height, body mass and body composition. Participants were then driven (a 7 min journey) to the assigned environment, where they completed an investigator-led 60 min walk along a pre-determined 4 km route, with a 15 min seated rest at the mid-way point. Participants were then driven back to the laboratory, where baseline measures were repeated. Eating behaviour was assessed using a blinded method; following the completion of the stress and mood measurements, participants were told they would need to wait an hour for a further set of measurements and were then offered a complementary lunch. The amount and type of food consumed was recorded by investigators.

### 2.3. Measurements

#### 2.3.1. Mood

Participants completed a printed version of the Abbreviated POMS questionnaire [54]. The POMS allows responses to be segmented into five negative affect sub-scores (tension, depression, anger, fatigue, and confusion) and two positive affect sub-scores (vigour and esteem-related affect [ERA]). The sub-scores were combined to generate an overall mood rating referred to as total mood disturbance (TMD) score: (sum of all negative affect scores − sum of all positive affect scores) + 100.

#### 2.3.2. Physiological Stress

RHR and BP were measured after a 10 min rest period by using a sphygmomanometer (M6, Omron, Kyoto, Japan) on the non-dominant upper arm while the patient was sitting upright. Saliva samples were collected via the passive drool method using Saliva Collection Aids (Salimetrics LLC., Ely, UK) and stored at −80 °C. Salivary cortisol concentrations were

assessed in duplicate according to the manufacturer's instructions using a commercially available ELISA (Salimetrics, LLC., Ely, UK). Cortisol samples with a coefficient of variation exceeding 10% were taken again.

### 2.3.3. Energy Intake and Ghrelin

As previously described, participants were blind to the measurement of energy intake. The participants were given a large bowl of cheese and tomato pasta (fusilli pasta, pecorino, tomato and chilli pasta sauce and olive oil; all Tesco, Welwyn Garden City, UK), along with a side of chocolate buttons (Tesco Milk Chocolate Buttons 70G, Welwyn Garden City, UK). The cheese and tomato pasta meal was homogenous in composition and provided 6.74 ($\pm$0.05 SD) kJ$\cdot$g$^{-1}$ of energy (with 14%, 60%, 24% and 2% of the energy provided by protein, carbohydrate, fat and fibre, respectively). One participant was vegan and was provided with a vegan equivalent [fusilli pasta, Violife Epic Mature Cheddar flavour (London, UK), tomato and chilli pasta sauce and olive oil; all Tesco except cheese alternative, Welwyn Garden City, UK], along with a side of vegan chocolate (70 g portion of Green & Black's Organic 70% Cocoa Intense Dark chocolate, Uxbridge, UK). The content of the vegan meal was similar in nutrient profile: 6.61 ($\pm$0.12 SD) kJ$\cdot$g$^{-1}$ (with 11%, 64%, 23% and 2% of the energy provided by protein, carbohydrate, fat and fibre, respectively). The location, serving bowl and cutlery were standardised. Energy intake was measured based on the difference in weight of the food before and after eating. The pasta was categorised as the 'savoury' option and the chocolate buttons were categorised as the 'sweet' option. Total energy intake, calories from savoury food, calories from sweet food, and the sweet:savoury ratio were used to analyse energy intake. Upon completion of the study, participants were informed of the blinded measure and provided written consent for these data to be reported.

Saliva samples were used to measure salivary ghrelin as an indicator of hormonal change in hunger signalling [55] using commercially available Elisa kits (Stratech, LLC., Ely, UK). Samples, run in duplicates, were re-run if the coefficient of variation exceeded 10%.

### 2.3.4. Other Measures

Prior to the first intervention, participants completed a general health questionnaire (Loughborough University health screening questionnaire) and an assessment of body composition (Seca Medical Body Composition Analyser, Naarden, The Netherlands). Saliva samples were also analysed for oestradiol content (ELISA, Salimetrics, LLC., Ely, UK). Salivary oestradiol was only analysed at baseline measures as this information was used as an indicator of hormonal status related to the menstrual cycle [56]. Samples with a coefficient of variation that exceeded 10% were repeated.

### 2.4. Statistical Analysis

Data analysis was conducted using the SPSS statistics package (Version 27.0 for Windows; Armonk, NY, USA). All variables were initially tested for normality and outliers ($+/-$ 2.5 SD) were excluded from analysis [57]. A 2 $\times$ 2 ANOVA was used to identify the main effect of environment (nature versus green urban), time (baseline versus post) and interactions between environment and time, regardless of normality [58]. Independent $t$ tests, or the Mann-Whitney U test for non-parametric data, were used to evaluate the order effect between intervention groups. Post hoc tests were run for variables that had a significant environment-to-time interaction. Paired $t$ tests (or the non-parametric equivalent, paired samples Wilcoxon test) were used to evaluate the following comparisons: baseline vs. post-nature intervention, baseline vs. post-green urban intervention, baseline nature vs. baseline green urban intervention and post-nature vs. post-green urban intervention. A Pearson correlation coefficient was calculated for HR, BP and salivary cortisol in order to assess the relationship between baseline values and changes from baseline. Significance was defined as $p < 0.05$.

## 3. Results

### 3.1. Participants

The length of the average interval between the nature and green urban interventions was 3.9 days (SD 1.5 days). No significant differences were found for the baseline salivary oestradiol concentration between the nature (M = 1.3 pg/mL, SD = 0.6) and green urban interventions (M = 1.3 pg/mL, SD = 0.5; z = 0.21, $p$ = 0.83). There were no differences in any of the baseline measures in any outcome variable (POMS scores, HR, SBP, DBP, salivary cortisol, and salivary ghrelin) when comparing the nature and green urban interventions ($t_{(21)} \leq 5.68$, $p \geq 0.077$; z $\leq 1.8$, $p \geq 0.072$). One participant opted out of the test meal.

### 3.2. Mood

There was a significant interaction between environment and time for ERA; $F_{(1,21)} = 5.47$, $p = 0.029$. Post hoc tests identified an elevated ERA after the nature intervention relative to the results taken after the green urban intervention; $t_{(21)} = 3.42$, $p = 0.001$ (Figure 2).

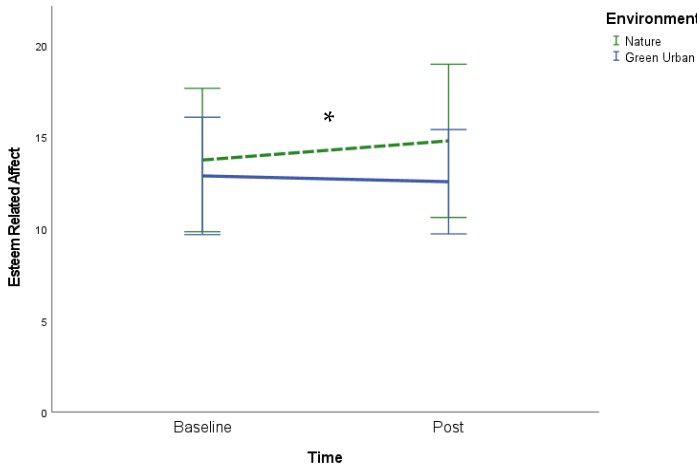

**Figure 2.** Esteem-Related Affect by Environment and Time. Mean $+/-$ standard deviation of baseline and post-intervention esteem-related affect. * Denotes significant interaction effect *n* = 22.

The interaction effect was not statistically significant for TMD or the subscales of tension, anger, fatigue, depression, confusion, and vigour; $F_{(1,21)} \geq 1.07$, $p \geq 0.313$. However, the main effect of environment was significant for TMD, depression, confusion, and ERA; $F_{(1,21)} \geq 4.98$, $p \leq 0.037$. TMD, depression, and confusion were higher following the green urban intervention, while ERA was higher following the nature visit. The main effect of time was significant for all POMS measures, $F_{(1,21)} \geq 5.69$, $p \leq 0.027$, except ERA; $F_{(1,21)} = 0.56$, $p = 0.46$. TMD, tension, anger, fatigue, depression, and confusion decreased with time, while ERA and vigour increased (Table 1).

**Table 1.** Profile of Mood States Results.

| | Nature | | Green Urban | |
|---|---|---|---|---|
| | **Baseline** | **Post** | **Baseline** | **Post** |
| TMD * | 91.0 (10.5) | 82.6 (9.0) | 95.4 (9.4) | 89.7 (9.0) |
| Tension | 2.6 (3.4) | 1.2 (1.7) | 2.7 (2.9) | 1.8 (2.3) |
| Anger | 1.0 (1.5) | 0.4 (0.8) | 1.1 (1.5) | 0.5 (1.3) |
| Fatigue | 3.9 (2.5) | 2.1 (2.1) | 4.6 (3.0) | 3.3 (2.9) |
| Depression * | 0.6 (1.1) | 0.2 (0.6) | 1.3 (1.8) | 0.9 (1.8) |
| Confusion * | 2.1 (1.9) | 0.8 (1.2) | 3.0 (2.9) | 1.4 (1.7) |
| Vigour | 5.4 (4.0) | 7.2 (4.3) | 4.5 (3.2) | 5.5 (2.8) |
| **ERA *** | 13.7 (3.9) | 14.8 (4.2) | 12.9 (3.2) | 12.6 (2.8) [†] |

Mean (standard deviation) reported; *n* = 22. Significant time-to-environment interaction is **in bold**; * statistically significant main effect of time; [†] statistically significant difference from post-nature. TMD = total mood disturbance, ERA = esteem-related affect).

### 3.3. Physiological Stress

The interaction effect between environment and time was not statistically significant for RHR, SBP, DBP or salivary cortisol; $F_{(1,21)} \leq 1.97$, $p \geq 0.18$. The main effect of environment was significant for RHR; $F_{(1,21)} = 6.15$, $p = 0.02$, with a higher RHR recorded in the nature intervention than the green urban visit. The main effect of time was significant for RHR, SBP, DBP and salivary cortisol; $F_{(1,21)} \geq 4.67$, $p \leq 0.04$, where all measures decreased after the intervention (Table 2). A significant negative correlation was observed between baseline cortisol levels and the magnitude of change in both the nature ($r_{(21)} = -0.71$, $p < 0.001$) and urban ($r_{(20)} = -0.91$, $p < 0.001$) group (Figure 3).

**Table 2.** Physiological stress results. All *n* = 22.

| | Nature | | Green Urban | |
|---|---|---|---|---|
| | **Baseline** | **Post** | **Baseline** | **Post** |
| Resting Heart Rate (beats/min) *^ | 63.3 (9.4) | 59.7 (9.3) | 60.6 (11.1) | 57.5 (9.8) |
| Systolic BP (mmHg) * | 108.7 (8.7) | 106.7 (8.6) | 109.7 (8.9) | 107.1 (9.3) |
| Diastolic BP (mmHg) * | 65.4 (5.4) | 64.6 (5.1) | 66.2 (7.5) | 65.0 (7.0) |
| Cortisol (nmol mL$^{-1}$) * | 14.0 (7.4) | 7.5 (5.4) | 16.1 (10.4) | 6.3 (3.2) |

Mean (standard deviation) reported. BP = blood pressure. n = 22. * Statistically significant main effect of time; ^ statistically significant main effect of environment.

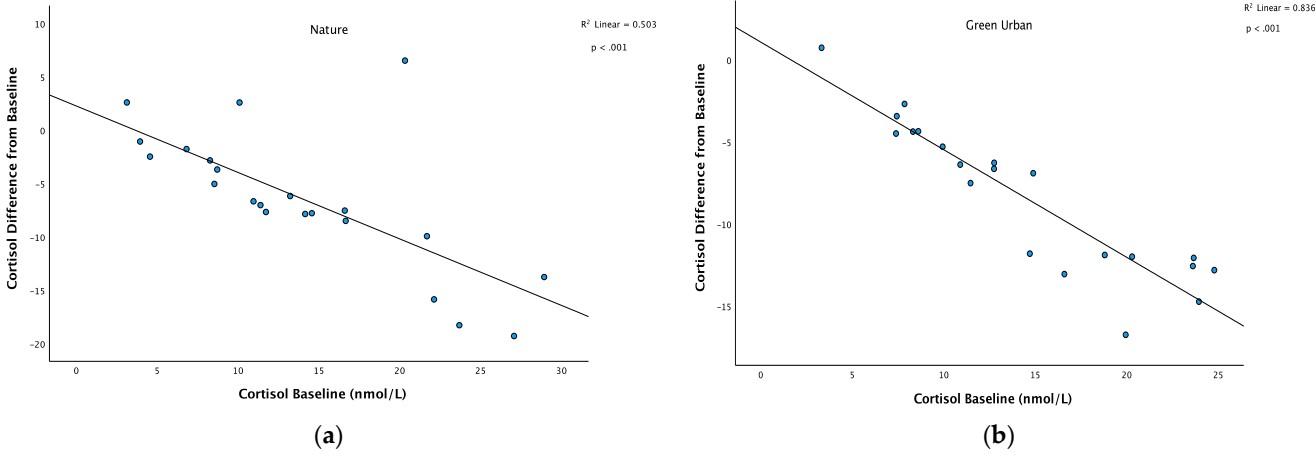

(**a**)  (**b**)

**Figure 3.** Correlation of Baseline Cortisol and Magnitude of Change. Cortisol: baseline changes correlation in nature (**a**) and green urban (**b**) intervention. *n* = 22.

### 3.4. Energy Intake and Ghrelin

There was no significant time-to-environment interaction for salivary ghrelin; $F_{(20)} = 0.229$, $p = 0.637$. While there was no main effect of environment, $F_{(1,21)} = 0.39$, $p = 0.54$, there was a main effect of time; $F_{(1,21)} = 9.76$, $p = 0.005$. There were no significant differences in energy intake (savoury, sweet or total, and the sweet:savoury ratio) following the walk in either environment (Table 3). Analyses was repeated both with and without the vegan participant and the results were unaffected.

**Table 3.** Eating Behaviour Results.

| | Nature | Green Urban | |
|---|---|---|---|
| | **Post** | **Post** | **Post-Nature vs. Post-Green Urban *t/z*** |
| Total energy intake (kJ) | 3138 (923) | 3206 (1037) | −0.54 [a] |
| Savoury (kJ) | 2401 (893) | 2446 (915) | −0.29 [a] |
| Sweet (kJ) | 740 (337) | 760 (405) | −0.78 [a] |
| Sweet:savoury ratio | 0.4 (0.6) | 0.4 (0.3) | −0.11 [b] |

Mean (standard deviation) reported. [a] paired samples *t* test [b] Wilcoxon Signed Rank Test, no significant results were reported; *n* = 21 (one participant declined to eat and another was an outlier [Below 2.5 SDs]).

## 4. Discussion

### 4.1. Benefits of Walking Outdoors

The observed physiological and psychological benefits associated with walking are consistent with those found in prior research. The positive relationship between physical activity and mood is well documented [27], and signs of benefit have been reported within 20 min of initiating exercise [59]. Although improvements in tension, depression, and anger have consistently been reported, reports of the effect of walking on fatigue have been less consistent [57–61]. This is likely due to variation in exercise intensity, affecting both the depletion of energy reserves and the opportunity to pay attention to the environment [62]. The present study found that light exercise outdoors reduces fatigue.

In addition to an improvement in mood, we report reductions in markers of physiological stress (heart rate, blood pressure and salivary cortisol) following both walks. This is consistent with the meta-analyses and longitudinal investigations that argue chronic physical activity decreases resting heart rate and blood pressure [61–63].

Previous studies that have considered the effect of environment in the absence of exercise on physiological stress have consistently reported reduced physiological stress in natural settings relative to urban settings [45,47,64–66]. In the present study, in which participants walked for 75 min in natural and green urban settings, there was no differential influence of the environment. This result runs contrary to the relative decrease in cortisol recently reported following a 15 min walk in nature [67,68]. However, our findings are consistent with those of a recent systematic review reporting no significant differences in heart rate when exercising with a view of nature versus a view of a city centre [44]. In the present study, the 2 h time difference between pre- and post-walk sampling likely drove the significant cortisol decrease due to cortisol's pronounced circadian rhythm [69].

A significant negative correlation was observed between baseline cortisol and change in cortisol following both walks, indicating a greater cortisol reduction in participants with higher baseline pre-walk cortisol. This finding is potentially valuable for public health interventions as it suggests that individuals with higher levels of baseline stress benefited the most from the walking intervention in both environments.

### 4.2. Additional Benefits of Walking in Nature

Walking in nature conferred additional benefits to mood that were not seen following the green urban walk. Specifically, TMD, depression, confusion and ERA were overall more optimistic in the nature intervention relative to the green urban walk. This is consistent with evolutionary theory underpinning the concept of environmental mismatch [68,69], and suggests that the presence of nature may provide an additive benefit to the well-documented positive impacts of physical activity for mental health [70–72]. An anticipatory effect, associated with positive associations with nature, may have contributed to the differences found in this study. A 2019 systematic review noted that participants with advance knowledge of their assigned environment had lower baseline salivary cortisol in nature compared to those undergoing urban intervention [73]. The results of the present study might also be interpreted to indicate that this anticipatory effect could also apply to mood outcomes. Perceptual changes associated with the two test environments indicate the need to further investigate the motivational process in the context of green exercise, an issue that has not previously been reported on [74].

It is worth noting that previous studies investigating the influence of natural and urban environments on human physiology and psychology have used busy city environments with an absence of any greenery [70,71,73]. In the present study, participants completed the "green urban" walk along a dual carriageway (UK Classification "A road") lined with unkept foliage such as large trees, bushes and weeds. While participants were mainly exposed to road traffic noise and views of urban-built contexts during the green urban intervention, they were not completely removed from natural elements.

An explanation as to why stress might be reduced in natural environments lies in our evolutionary past. Much of the human evolutionary journey took place in predominantly

natural environments. Since the first emergence of life on Earth some 4 billion years ago [75,76], the appearance of the hominin lineage 6–7 million years ago [77] and the arrival of anatomically modern humans ~300,000 years ago [78], nature's forests, plains, coastlines, and mountains have provided the main environmental parameters within which natural selection acts to shape our biology. Consequently, due to the selection pressures imposed in those natural ancestral habitats, contemporary human biology is primarily adapted to those natural environments. By contrast, there has been a rapid and significant change in the primary habitat of our species over the last 200–300 years. During this time—a fraction of the human evolutionary timeline—the majority of the human population worldwide has migrated from rural areas to urban centres [77,78]. Contemporary cities provide few of the natural features that characterised the ancestral environments that shaped our species' evolution. Additionally, they pose a novel set of environmental stressors associated with sensory stimulation via artificial noise and light, as well as with air pollution [30,79,80].

Longman and Shaw recently proposed that the speed and the severity with which our habitat has changed has created an environmental mismatch, arising from the discordance between the natural environments to which our biology is primarily adapted and the cities that most of our species today calls home [68,69]. The deleterious health consequences of the environmental mismatch can be seen in both the aforementioned experimental work, describing the stress-relieving effects of exposure to nature [31,32], and the high levels of chronic stress globally associated with living in urban centres [1,30,79] (see [47,48] for more detail).

### 4.3. Eating Behaviour

No differences were observed between the two interventions in terms of salivary ghrelin or eating behaviour. While prior investigations have failed to establish a consistent association between exercise and ghrelin levels [81–83], low-intensity exercise such as walking is not considered to have a suppressive effect on ghrelin [84,85]. Hunger and stress are reciprocally linked as high levels of circulating ghrelin stimulate the release of cortisol, and stress is associated with increased energy intake and decreased sensitivity to leptin signalling [14,83]. In the present study, both environments led to a decrease in markers of physiological stress and a reduction in ghrelin. Further studies are required to disentangle the effects of exercise and environment on appetite signalling and eating behaviour.

### 4.4. Limitation

The sample size of this study ($n = 22$) was small. This limited the investigation's statistical power and our ability to detect small (but potentially meaningful) changes in measures such as eating behaviour between environments. Furthermore, and perhaps most influentially, the urban environment was not devoid of natural features, which limited potential contrast between the two intervention environments. It is possible that this reduced the effect on the variables under investigation. The study may therefore have benefited from the inclusion of a third intervention group in an urban environment lacking any natural features. Future work that includes locations which are representative of the continuous spectrum of greenery in urban and natural environments may provide further insights into the dosing effect of nature. Lastly, this study measured the immediate effect of physical activity in two natural conditions. Longitudinal analysis is required to understand the cumulative effects of a longer exposure period.

### 5. Conclusions & Implications

We found walking in both green urban and natural environments to be beneficial for mood and physiological stress and observed that walking in nature conferred additional advantages for mood (as indicated by the time-to-environment interaction in esteem-related affect). The main effects of environment in terms of positive and negative affect suggest that there may be an anticipatory effect of nature exposure, whereby attitudes toward nature amplify the influence of the environment on mood. These preliminary findings have

important implications for public health. First, our results provide evidence in support of the use of mild exercise to benefit mental health, irrespective of environment. Second, this study demonstrates that the utilisation of green spaces for mild exercise, particularly forests, can facilitate some additional mental health benefits. Further work, featuring a high-powered longitudinal analysis of the effects of repeated exposure to physical exercise in nature, may provide more insight regarding the impact of exercise in natural environments on markers of health and wellbeing.

**Supplementary Materials:** The following supporting information can be downloaded at: https://www.mdpi.com/article/10.3390/urbansci7030085/s1, Supplemental Information S1: Description of Burleigh Woods.

**Author Contributions:** Conceptualization, Y.T., D.P.L. and C.S.; methodology, Y.T., E.O., A.B., N.B., L.J., D.P.L. and C.S.; formal analysis, Y.T.; writing—original draft preparation, Y.T.; writing—review and editing, Y.T, D.P.L., C.S., L.J. and N.B.; visualization, Y.T.; supervision, L.J., N.B., D.P.L. and C.S.; project administration, Y.T., I.W. and H.T.; funding acquisition, Y.T. All authors have read and agreed to the published version of the manuscript.

**Funding:** This research was funded by the Society for the Study of Human Biology (SSHB 2021-01).

**Data Availability Statement:** The data presented in this study are available on request from the corresponding author. The data are not publicly available due to ethical restrictions.

**Acknowledgments:** Thank you to Matthew Roberts and Callum Mould at Loughborough University for providing technical training and supporting data collection and analysis.

**Conflicts of Interest:** The authors declare no conflict of interest. The funders had no role in the design of the study; in the collection, analyses, or interpretation of data; in the writing of the manuscript; or in the decision to publish the results.

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
