# Peer review of "Additional Health Benefits Observed following a Nature Walk Compared to a Green Urban Walk in Healthy Females"

_urbansci, doi:10.3390/urbansci7030085_

Round 1

Reviewer 1 Report

I would like to thank for the opportunity to review this manuscript on an important topic. Overall, it is a well written manuscript. However, as with most of the submissions, I have also some suggestions to further improve the quality of this manuscript.

In the abstract, Authors could add some more specific values to support their results. Also, some sentence for the conclusions is also recommended to add.

In the introduction, Authors give a great overview of physical activity and expose to nature benefits. However, Authors miss the literature review of which kind of environments support the higher levels of physical activity? For example, environments that provide autonomy support as well as are intrinsically motivating can support one’s physical activity (for example, see a study by Kalajas-Tilga et al. 2020).

Kalajas-Tilga, H., Koka, A., Hein, V., Tilga, H., & Raudsepp, L. (2020). Motivational processes in physical education and objectively measured physical activity among adolescents. Journal of Sport and Health Science, 9(5), 462–471. https://doi.org/10.1016/j.jshs.2019.06.001

Tables and figures could be modified. Please check their accordance with journal guidelines.

The discussion lacks practical implications of this study. Please add some specific recommendations based on the results of this study.

Also, Authors are highly recommended to add some specific suggestions for future research and future interventions. For example, future intervention studies would greatly benefit by relying of classification of motivational behaviours by Ahmadi et al. (2023) to support one’s intrinsic motivation towards physical activity as well as to support exposure to nature benefits.

Ahmadi, A., Noetel, M., Parker, P., Ryan, R. M., Ntoumanis, N., Reeve, J., Beauchamp, M., Dicke, T., Yeung, A., Ahmadi, M., Bartholomew, K., Chiu, T. K. F., Curran, T., Erturan, G., Flunger, B., Frederick, C., Froiland, J. M., González-Cutre, D., Haerens, L., . . . Lonsdale, C. (2023). A classification system for teachers’ motivational behaviors recommended in self-determination theory interventions. Journal of Educational Psychology. Advance online publication. https://doi.org/10.1037/edu0000783

Please check for grammar.

Reviewer 2 Report

Dear authors,

Thank you for giving me the opportunity to review this submitted manuscript.

As a tip, I suggest you include the word "women" in the title of your article.

This is an interesting study. In general, the document is well written and structured. Very few grammar corrections need to be adjusted. In addition, it was well founded, with adequate analyzes and met the objective proposed by the authors. However, in my opinion, the document should address:

-I suggest that throughout the article you use the acronym Physical Activity (PA).

-In short, what is TMD? I suggest you not put the acronym in until its meaning has been explained.

-should have latest scientific literature in the introduction

-What are the limitations of the study? It would be interesting to mention them.

Reviewer 3 Report

The study presented for review is relevant both from the point of view of determining the impact of various environmental conditions on the human body and identifying new forms of physical activity in the fight against stress and obesity in mature women. Among the advantages of the presented study is an innovative vision of the widespread use of instrumental research methods in assessing the impact of physical activity on the body of the subjects.

The terminological apparatus of the study requires clarification and presentation of the author's vision (‘green exercise’).

In describing the relevance of the study, the author focuses on explaining the need for research from a variety of perspectives, while more attention should be paid to information on similar studies and a clear list of unresolved issues. The author's statements [41] about the low quality of previous studies require additional explanation.

The general references to the works of other authors are mostly old (more than 5 years), which is 67% of the total number of references. It is also necessary to find out the previous experience of the authors of the study on the topic under consideration, since only two of the authors have joint publications presented in the list of references.

The manuscript is scientifically sound and the proposed model of the pedagogical experiment is acceptable to confirm the hypothesis. At the same time, it should be specified that this is the immediate effect of physical activity in certain natural conditions and the analysis of the cumulative effect of longer exposure would have made the study more fundamental.

The applied research methods fully allow us to reproduce the pedagogical experiment. Additional explanation is needed to determine the number of subjects in each group and the subsequent reduction in the number of subjects in the analysis of the results (for example, the analysis of Profile of Mood States Results n=22) and whether there was interaction between participants in each group during the walk.

The data presented in the tables and figures need to be revised to demonstrate statistically significant differences in the indicators at the beginning and end of the experiment. The presentation of numerical data in terms of mean and standard deviation needs to be revised, in some cases, the standard deviation is 2, 3 times higher than the mean (Table 1. Profile of Mood States Results; Table 2. Physiological stress results in Cortisol (nmol mL-1). In our opinion, the description should be applied to data that do not follow the normal distribution law of Me (25;75). From the point of view of visual perception of numerical data, the data presented in Figures 2, 3, 4, and 6 do not require display in the form of figures; their description in the text part of the article is sufficient.

In the text of the article, I would like to see the author's explanation of the decrease in the level against the background of physical activity.

There is no reference to table 2 in the text.

The conclusions of the paper reflect the results of the study, but at the same time, they need to be supplemented with the results of statistical analysis and specific numerical data.

Round 2

Reviewer 1 Report

Authors have done well job on revising the manuscript.